# Diagnosis Biomarkers of Cholangiocarcinoma in Human Bile: An Evidence-Based Study

**DOI:** 10.3390/cancers14163921

**Published:** 2022-08-13

**Authors:** Fang Bao, Jiayue Liu, Haiyang Chen, Lu Miao, Zhaochao Xu, Guixin Zhang

**Affiliations:** 1Institute of Integrative Medicine, Dalian Medical University, No. 9, South Road of Lvshun, Dalian 116044, China; 2Department of General Surgery, Pancreatic-Biliary Center, The First Affiliated Hospital of Dalian Medical University, No. 222, Zhongshan Road, Dalian 116011, China; 3CAS Key Laboratory of Separation Science for Analytical Chemistry, Dalian Institute of Chemical Physics, Chinese Academy of Sciences, 457 Zhongshan Road, Dalian 116023, China

**Keywords:** cholangiocarcinoma, bile, diagnosis, biomarker

## Abstract

**Simple Summary:**

A liquid biopsy has the characteristics of low trauma and easy acquisition in the diagnosis of cholangiocarcinoma. Many researchers try to find diagnostic or prognostic biomarkers of CCA through blood, urine, bile and other body fluids. Due to the close proximity of bile to the lesion and the stable nature, bile gradually comes into people’s view. The evaluation of human bile diagnostic biomarkers is not only to the benefit of screening more suitable clinical markers but also of exploring the pathological changes of the disease.

**Abstract:**

Cholangiocarcinoma (CCA) is a multifactorial malignant tumor of the biliary tract, and the incidence of CCA is increasing in recent years. At present, the diagnosis of CCA mainly depends on imaging and invasive examination, with limited specificity and sensitivity and late detection. The early diagnosis of CCA always faces the dilemma of lacking specific diagnostic biomarkers. Non-invasive methods to assess the degree of CAA have been developed throughout the last decades. Among the many specimens looking for CCA biomarkers, bile has gotten a lot of attention lately. This paper mainly summarizes the recent developments in the current research on the diagnostic biomarkers for CCA in human bile at the levels of the gene, protein, metabolite, extracellular vesicles and volatile organic compounds.

## 1. Introduction

Cholangiocarcinoma (CCA) is a type of hepatobiliary malignant tumor that is highly deadly and aggressive, and its prevalence is increasing year after year [1]. Based on anatomical location, CCAs are classified as intrahepatic CCA (iCCA), perihilar CCA (pCCA) and distal CCA (dCCA). Extrahepatic CCA refers to pCCA and dCCA, which are also known as extrahepatic CCA (eCCA) in epidemiology [2]. Since CCA in some regions, such as South-East Asia is mainly related to liver fluke, the epidemiology of CCA can be categorized as fluke-related and non-fluke-related CCA [3]. The main risk factors of CCA are associated with bile stasis and chronic inflammation of the biliary epithelium [2]. Mortality from iCCA varies greatly across regions, but in most countries, it increased from less than 1/10,000 in the early 2000s to 1–2/100,000 in 2014, which was closely related to the frequency of risk factors and diagnostic difficulties caused by unobvious early symptoms, whereas mortality from eCCA has decreased with the increased use of laparoscopic cholecystectomy [4]. As a result, enhancing the accuracy of early CCA diagnosis, in addition to minimizing risk factors, is another method to reduce CCA mortality.

At present, the commonly used clinical imaging methods such as CT or MRI cannot determine whether the biliary pathology is malignant. Furthermore, the diagnostic specificity and sensitivity of brush cytology and fluorescence in situ hybridization (FISH) analysis by encoscopic retrograde cholangio-pancreatography (ERCP) is low [5,6]. Singhi et al. proposed a multigene a next-generation sequencing (NGS) diagnosis method in tissues, which was significantly more effective than serum CA19-9 and pathological evaluation. However, it has not been included in the management guidelines of CCA [7].

Bile secreted by the liver cells, generally stored in the gall bladder and discharged to the duodenum in the digestive process, is an important part of the biliary system. At present, the main methods to obtain bile are duodenal drainage, gallbladder puncture and direct surgery, among which surgical methods include ERCP and cholecystectomy. Moreover, the bile collected by duodenal drainage contains part of gastric juice and pancreatic juice, which is not conducive to the purification and analysis of bile. Bile samples should be transported on ice after acquisition and frozen at −80 °C until use. For different types of bile candidate markers, treatment and analysis methods are as follows: (1) Gene level: After extracting DNA from the bile, amplification of the target fragment by PCR to obtain the expression of potential diagnostic biomarkers and analysis of data [8]. For RNA, target RNA reverse transcription is required to form a stable CDNA before PCR amplification and data analysis, and real-time quantitative PCR is usually used [9,10]. (2) Protein level: enzyme linked immunosorbent assay (ELISA) or automatic immunofluorescence analysis is adopted to verify suspicious protein biomarkers in bile, but when exploring unknown diagnostic biomarkers, proteomics methods are being generally used to find markers and then checked by ELISA in bile [11,12,13]. (3) Metabolism level: for bile metabolites, the diagnostic efficacy of suspected biomarkers is generally analyzed by magnetic resonance spectroscopy or quantitated by liquid chromatography tandem mass spectrometry [14,15]. (4) VOC is mainly through heating the sealed bottle containing bile and extracting the air above, and the collected gas is analyzed by ion flow tube mass spectrometry so as to identify the markers and analyze the diagnostic effect [16].

Some researchers are trying to find novel diagnostic biomarkers for CCA in human body fluids such as serum or urine [17,18]. Among them, bile with a smaller metabolic range is easier to reliably identify biliary disease in comparison to serum, which is susceptible to other organ diseases or complications. On the other hand, higher concentrations of tumor biomarkers may be present in bile because the bile is in direct contact with the lesion, which makes it easier to detect. The consistency of gene mutations between tumor tissues and bile was twice that in serum, and serum had the lowest levels of ctDNA [19]. Another study also proved that bile ctDNA is superior to ERCP cytology [20]. Nowadays, a growing number of studies have aimed to discover candidate biomarkers of CCA with strong diagnostic ability in bile. We differentiated the available results by gene, protein, compound, EV source, VOS source and validation method (Figure 1). The current state of diagnosis biomarkers research in CCA utilizing bile as a sample will be summarized in this review.

## 2. Search for CCA Diagnosis Biomarkers in Human Bile at Gene Level

Tumors originate from the uncontrolled cell proliferation caused by gene mutations. Diagnosing tumors at the genetic level is conducive to the early treatment and prevention of the progression of malignant transformation. Cell-free DNA (cfDNA) in bile is about 20 times that in serum, and its mutation spectrum can be used to diagnose malignant biliary strictures. Its diagnostic performance is much higher than that of serum CA19-9, which further proves the potential of bile to diagnose CCA [21]. Currently, DNA methylation and microRNA (miRNA) are the main directions for finding diagnostic biomarkers for CCA at the level of bile genes.

### 2.1. DNA Methylation and CCA Diagnosis Biomarkers

DNA methylation is a critical epigenetic regulatory mechanism that can add methyl groups to DNA molecules to influence DNA activity without changing the DNA sequence. Abnormal DNA methylation is one of the common epigenetic signs in human cancer and is closely related to tumor development and metastasis [22].

#### 2.1.1. p16INK4a and p14ARF Promoter Methylation

p16INK4a is one of the four isoforms of p16 transcription, which is a tumor suppressor protein that acts as a negative regulator of the proliferation of normal cells that induce g1 cell cycle arrest by interacting strongly with CDK4 and CDK6 [23]. p16 is one of the possible biomarkers for assessing cervical lymph node metastasis in oral squamous cell carcinoma, and an NGS study of 26 patients with cholangitis-associated cholangiocarcinoma (PSC-CCA) or primary biliary cholangitis (PSC) showed the aberrant expression of p16 consistent with heterogeneity in PSC-CCA [24,25]. Moreover, this NGS result also showed the potential of p16 to distinguish between PSC-CC and PSC. Inactivation of the p16 gene is a frequent event in CCA, and the main mechanism is the promoter methylation of the p16 gene [26]. p14ARF binding to MDM2 can cause cell cycle arrest through activation of the P53 response [27]. The methylation of p16 and p14 promoters in the bile of CCA patients is highly consistent with biliary malignancy compared with patients with benign biliary tract disease, implying that p16INK4a and p14ARF promoter methylation status is a candidate biomarker for the endoscopic diagnosis of biliary tract diseases, although the combined diagnosis of CCA is not sensitive. Additionally, p16INK4a showed a sensitivity of 53.6% and a specificity of 93.8% for discriminating between CCA and benign illnesses, whereas p14ARF had a sensitivity of 46.2% and a specificity of 96.9% [8]. The study of Zhang et al. established a promising biomarker for CCA with a specific bile gene methylation group consisting of six genes, including P16INK4a [28]. This suggests that p16INK4a and p14ARF are promising biomarkers for the diagnosis of CCA, but issues of reduced sensitivity, such as the combination of additional biomarkers for diagnosis, must be addressed.

#### 2.1.2. Combined Diagnosis of Different Gene Methylation

Shin et al. integrated 59 methylation indicators from the DNA methylation study of eCCA tissues to verify and evaluate them in 77 bile specimens and finally found a five-gene panel composed of CCND2, CDH13, GRIN2B, RUNX3 and TWIST1 that can be used to distinguish CCA patients from benign groups in bile with 75.6% sensitivity and 100% specificity and which also can supple 39% of biliary cytological missed cases [9]. Droplet digital PCR (ddPCR) can absolutely quantify DNA methylation in samples. As absolute quantification of DNA methylation in samples was possible by ddPCR; Vedeld et al. analyzed 344 bile samples containing CCA and other benign biliary diseases by ddPCR, indicating the sensitivity and specificity of combing diagnosis of CDO1, CNRIP1, SEPT9 and VIM promoter methylation were 100% and 90%, respectively [10]. The earliest incidence of tumor growth is generally genetic variation, and methylight technology is reasonably developed and provides advantages in the detection of early CCA. CCA cannot be diagnosed clinically by detecting a single gene variant. These results show that the combined diagnosis of multi-gene variants can improve diagnostic efficiency, although more data is required [9,10].

When compared to single DNA methylation or serum CA19-9, the combined diagnosis of diverse gene methylation has a higher sensitivity and specificity, and it can also reduce missed diagnosis rates by broadening diagnostic panels. However, existing technology as well as the expense of expanding diagnostic panels must be addressed. We should do everything possible to increase the rate of diagnosis while keeping the price low and the diagnostic time short.

#### 2.1.3. Detection of DNA Methylation Biomarkers in Bile

DNA methylation in bile as a diagnostic biomarker for CCA is feasible, but we are facing a dilemma about how to achieve a simple and rapid detection. Liquid chromatography, which combines fluorescence, UV and mass spectrometry, is an early and sensitive method for the detection of genome-wide DNA methylation, but it cannot obtain the methylation location [29]. Subsequently, PCR and electrophoresis techniques were used to detect DNA methylation; however, due to heavy workload and low resolution, this method was gradually replaced by microarray technology and sequencing, both of which have the advantages of high throughput and high resolution [30,31]. Recently, many studies attempted to use fluorescence sensors and electrochemical sensors to detect DNA methylation in vivo or in vitro. Among them, fluorescence sensors, which are based on fluorescence probes and DNA chain exchange, have a high sensitivity, low consumption, easy operation and advantages in real-time detection [32]. Therefore, selecting DNA methylation biomarkers or combined methylation biomarkers in human bile and searching for specific fluorescent labeling methods and calculation methods are one of the good methods for the clinical application of DNA methylation biomarkers.

### 2.2. MiRNA and CCA Diagnostic Biomarkers

MiRNA is an endogenous non-coding, short-stranded RNA with a length of 19–25 nucleotides that can regulate gene expression by binding to the target gene. MiRNA can be divided into body fluids miRNA (BF-miRNA) and tissue miRNA (T-miRNA). Studies have found a significant positive relationship between BF-miRNA and T-miRNA in health or disease states, suggesting that BF-miRNA, like T-miRNA, can also be a biomarker for some diseases [33]. MiRNA regulates cell formation, proliferation, apoptosis and autophagy, and its expression level is a diagnostic biomarker for a range of disorders, including cancers, with the potential to forecast early disease and assist therapy [34,35,36,37,38,39,40,41,42,43,44]. MiRNA exerts on the secretion, apoptosis, proliferation and migration of cholangiocytes under regulated mechanisms and conditions such as epigenetic, hypoxia or circadian rhythms [45]. Small RNA library sequencing and bile fractionation confirmed that endogenous miRNAs in bile was mainly distributed in cells and nuclei, and that remained stable under denaturing conditions such as in strong acids and strong bases [46]. Although the stability of miRNA is stronger in plasma, the diagnostic effectiveness of miRNA in bile is remarkably higher than in body liquids samples such as plasma and urine [47,48].

#### 2.2.1. MiR-9

MiR-9 is a tumor gene with a high expression and long-term stable existence in the bile of CCA [46]. Many studies believe that the abnormal expression of the miR-9 family is related to the development and metastasis of a variety of tumors including hepatocellular carcinoma and ovarian cancer [49,50,51,52,53,54]. Compared with the bile of patients with benign biliary diseases, ten miRNAs (miR-9, miR-145*, miR-105, miR-147b, LET-7F-2*, let-7i*, miR-302c*, miR-199a-3p, miR-222* and miR-942) were significantly overexpressed in the bile of patients with CCA, among which miR-9 emerged as the most potential biomarker due to its diagnostic sensitivity and specificity of 88.9% and 100%, respectively [46].

#### 2.2.2. RNU2-1f

Baraniskin et al. showed that the best discrimination value of miRNA is fragmented U2 snRNA, and U2 snRNA (RNU2) together with several proteins can form U2 small nuclear ribonucleoprotein (snRNP), which plays a key role in splicing pre-mRNA catalyzed by the spliceosome [55]. RNU2-1 fragments (RNU2-1f) in circulating body liquids have been confirmed for the potential to be diagnostic markers in many kinds of tumors, such as pancreatic and colorectal adenocarcinoma, CNS lymphoma, and epithelial ovarian cancer and lung cancer [55,56,57,58,59]. Based on the above studies, Baraniskin et al. investigated the levels of RNU2-1f in bile in patients with CCA and benign biliary disease and found that RNU2-1f could also serve as a diagnostic biomarker of CCA in bile. However, RNU-1f is not yet available in clinical practice [60].

#### 2.2.3. Others

It has been proposed that miRNA profiles in serum and bile differ, and in bile miRNA profile, miR-640, miR-640, miR-1537 and miR-3189 have the potential to distinguish between PSC and PSC-CCA [54]. MiR-1537 combined with magnetic resonance imaging can be used for breast cancer diagnosis and prognosis evaluation [61]. MiR-412 in saliva has the potential to diagnose oral cancer, and in serum can distinguish toxic liver injury from ischemic liver injury [62,63]. MiR-640 differentially expresses in the different genetic subtypes of chronic lymphocytic leukemia [64]. Endogenous miR-3189 is a tumor suppressor that inhibits the expression of a large number of genes involved in regulating cell cycle and cell survival [65]. Though these miRNAs have the possibility to divide CCA and PSC, the sensitivity of them is too low to use in patients.

MiR-30d-5p and miR-92a-3p levels in bile were considerably greater in CCA patients than in those with benign biliary illness [66]. In comparison to CA19-9 and CEA in serum and miR-92a-3p in bile, miR-30d-5p in bile has the greatest potential for use as a clinical diagnosis biomarker of CCA [66]. MiR-30d-5p levels in urine exosomes also can assess the rate of progression of autosomal dominant polycystic kidney disease [67]. However, with a specificity of 60.5%, it may have difficulty distinguishing CCA from PSC, resulting in a high rate of misdiagnosis.

In the study of searching for diagnostic biomarkers in human bile of CCA, it is not only required to diagnose CCA at an early stage for early treatment to enhance the survival rate, but it is also necessary to distinguish CCA from other diseases such as PSC and pancreatic cancer. Among the existing research results, miR-9 has the best diagnostic performance, but it is imperative that it be verified in a larger sample range.

#### 2.2.4. Detection of miRNA Biomarkers in Bile

The potential of miRNA in body fluid biopsies of cancer patients is gradually being discovered. The earliest detection methods of miRNA mainly relied on Northern blotting analysis, but this method had low flux, low sensitivity and was relatively time consuming [68]. Microarray technology makes up for the shortcomings of low throughput and high cost, but the labeling method and probe design of miRNA detection still need to be improved [69,70]. qRT-PCR gradually become the gold standard for liquid miRNA diagnosis, but the low concentration of miRNA in liquid leads to a large error detection result [71]. Biosensors are reusable, efficient and fast [72]. In the future, combining fluorescence probe labeling with isothermal amplification technology to prepare sensors may be able to solve the problem of low miRNA concentration and complex background in liquid, as well as to detect specific miRNA or perform multi-channel multiplexing detection.

This section may be divided by subheadings. It should provide a concise and precise description of the experimental results, their interpretation, as well as the experimental conclusions that can be drawn.

## 3. Search for CCA Diagnosis Biomarkers in Human Bile at Protein Level

Protein is the material basis of human vital movement; abnormal protein expression and modification are also the inevitable ways of tumor formation and development. With the development of proteomic-related technologies, researchers began to construct tumor protein maps to reveal the intrinsic pathogenesis of tumors and the targets of tumors for diagnosis and treatment. This section will describe human bile proteins that may serve as diagnostic biomarkers of CCA based on different methods, such as ELISA, proteomics and glycoproteomics.

### 3.1. Diagnosis Biomarkers in Human Bile Proteins Based on ELISA

Increasing tumors biomarkers are being discovered; many investigators sought to validate in human bile whether biomarkers reported for diagnosing other tumors can be used to identify CCA and benign biliary tract disease, finding that Heat shock proteins 27 (HSP27), Heat shock proteins 70 (HSP70), Pyruvate kinase M2 (PKM2), sB7-H4, 11 ligand-binding repeats (sLR11) and minichromosome maintenance protein 5 (Mcm-5), which were differentially expressed in bile of patients with CCA and benign biliary tract disease, have the possibility of diagnosis of CCA.

#### 3.1.1. Heat Shock Proteins

Heat shock proteins (HSP) is a family of proteins containing multiple isoforms that are induced by high temperature or other emergency factors, playing a role in tumor formation and metastasis by involving in protein folding and maturation, and is also a potential tumor diagnostic biomarker and therapeutic target [73]. HSP27 in tissues can be utilized to predict the prognosis of iCCA, and it is also a potential biomarker for pancreatic cancer in serum [74,75]. HSP27 and HSP70 in bile are well in the diagnosis of iCCA, and the combined diagnostic pattern of HSP27 and HSP70 produced optimal sensitivity (90%) and specificity (100%) [13].

#### 3.1.2. PKM2

PKM2 is the M2 isoform of pyruvate kinase, whose aberrant expression and post-translational modifications can affect glycolysis, gene transcription, enzymatic activity and redox reactions in cancer, which have a negative effect on cancer [76]. PKM2 in blood has been proposed to screen for colorectal cancer, which is also a candidate marker for bladder and pancreatic cancer [77,78,79]. Malignant biliary stenosis contains non-CCA diseases such as pancreatic cancer. PKM2 in human bile can be utilized to identify malignant biliary stenosis, but it fails to distinguish CCA from other malignant biliary diseases, and its low diagnostic sensitivity restricts the ability to serve as a diagnostic biomarker [11].

#### 3.1.3. sLR11

A type I membrane protein, 11 ligand-binding repeats (sLR11) can be serum biomakers for the survival of diffuse large B-cell lymphoma or prognosis of non-Hodgkin’s lymphoma [80,81]. The expression of sLR11 is significantly increased in bile with biliary tract and pancreas tumors patients, and its capability to be a valuable diagnose biomarker is stronger than serum CEA and CA19-9 [82]. Further studies are needed to verify the role of sLR11 in CCA and the ability to distinguish between CCA and other malignant biliary stenosis diseases.

#### 3.1.4. sB7-H4

Soluble B7-H4 (sB7-H4) is a member of the tumor immune evasion mechanism, not only involved in the progression of multiple tumors but also a biomarker of tumors diagnosis and treatment [83]. One study found that B7-H2 was highly expressed in bile in CCA patients, with the possibility of diagnostic CCA, and the combined diagnosis with transpapillary forceps biopsy (ETFB) could also compensate for the shortcoming of insufficient diagnostic specificity of a single B7-H2 [84]. However, we cannot determine whether sB7-H4 in bile is produced by diseased bile duct tissue.

#### 3.1.5. Mcm-5

Minichromosome maintenance protein 5(Mcm-5) is a member of minichromosome maintenance family proteins, which are involved in DNA replication and cell proliferation [85]. The automatic immunofluorescence detection of Mcm-5 in urine or gastric aspirates is superior in detecting genito-urinary tract cancer and esophageal cancer [86,87]. Ayaru et al. compared the ability of mcm-5 in bile and biliary brush cytology to identify pancreatic bile duct malignancies, finding that while the diagnostic efficiency of bile Mcm-5 is higher than that of biliary brush cytology, its sensitivity rejects mcm-5 as a diagnostic biomarker of CCA [12].

These result show that HSP27 and HSP70 in human bile have the most advantages in diagnosing CCA; however, as the researchers only collected 20 bile samples, they need more clinical cases to verify the role of them.

### 3.2. Diagnosis Biomarkers in Human Bile Proteins Based on Proteomics

Proteomics is a method of analyzing proteins on a large scale, which can not only be used to discuss the process of disease occurrence and development, but also to search for diagnostic or prognostic biomarkers of tumors and other diseases by differential proteomics [88,89,90]. Rather than verifying diagnostic biomarkers in bile that have been reported in other diseases, proteomic techniques identify bile proteins that are most likely to be diagnostic biomarkers. It has been shown that proteomics is expected to be a means to diagnose or screen for CCA [91]. This part will introduce diagnostic biomarkers of CCA identified by proteomics in bile.

#### 3.2.1. Mac-2BP

Mac-2binding protein (Mac-2BP) plays a role in cell–cell and cell–matrix adhesion by binding to Galectin-1, Galectin-3, fibronectin and collagen [92], which is a candidate biomarker in nonalcoholic fatty liver disease diagnosis and prostate cancer diagnosis [93,94]. Jens et al. found high Mac-2BP expression in the bile mass spectrometry in biliary tract carcinoma patients, then measured Mac-2BP concentrations in bile and serum in 26 biliary cancer patients and 49 patients with benign biliary disease, and showed that the AUC of Mac-2BP was 0.7 to identify benign and malignant biliary diseases, almost identical to bile CA19-9; additionally, combined CA19-9 in bile with Mac-2BP significantly increased its AUC value to 0.75, while Mac-2BP in serum has no diagnosis value [95].

#### 3.2.2. CEAM6

Cell adhesion molecule 6 (CEAM6) takes part in uncontrolled proliferation and apoptosis, angiogenesis, immune evasion, etc., in the cancer process [96]. Differential proteomic analysis was performed on bile samples collected from patients with benign and malignant biliary stricture, and the results showed that CEAM6 was considerably expressed in the bile of patients with malignant biliary stricture [97,98]. CEAM6, a glycoprotein found in bile, can be employed as a possible biomarker for the detection of malignant biliary strictures after validation [97]. Another study also verified the potential of CEACAM6 to distinguish CCA from benign biliary tract disease in bile, but the diagnostic specificities of the two studies were 0.93 and 0.69, respectively [99]. The two more divergent results may be due to the former study that included non-CCA malignant biliary diseases, such as pancreatic cancer. This suggests that CEAM6 is unreliable in identifying CCA.

#### 3.2.3. NGAL/MMP-9 Complex

Neutrophil gelatinase associated lipocalin (NGAL) is a multifunctional protein released by neutrophils that plays a role in inflammation, immune function and carcinogenesis [100]. Zabron et al. collected bile during ERCP in 16 patients with malignant pancreaticobiliary disease and 22 patients with benign biliary tract disease and performed marker-free proteomic analysis, combined with Western blot analysis and ELISA analysis, finding that bile NGAL could distinguish between malignant pancreaticobiliary disease and benign biliary stenosis [101]. Udzinska et al. also believe that NGAL has good diagnostic potential in pancreatobiliary cancer [102]. However, there are no reports of NGAL used to diagnose a single CCA disease. MMP-9 is a member of the gelatinase subgroup of the MMP family, and TIMP-1 is an endogenous repressor of MMP-9. They have been reported as diagnostic markers of the giant cell tumor of bone, non-small cell lung cancer, cervical cancer, ovarian cancer, pancreatic cancer, osteosarcoma, breast cancer, colorectal cancer and gastric cancer [103,104,105,106,107,108,109,110]. A study found that TIMP-1 and MMP-9 are differentially expressed in the bile of patients with CCA and choledocholithiasis, but they have low diagnostic specificity and accuracy and, thus, are possible but less likely to serve as a diagnostic marker of CCA [111]. NGAL can form an NGAL/MMP-9 complex with MMP-9 to improve the stability of MMP-9. NGAL/MMP-9 complex in urine is a candidate marker for breast cancer, glioma and gastric cancer [112,113,114]. The high specificity of the complex can complement the deficiency of individual NGAL and MMP-9 in diagnosing CCA. Therefore, it is an interesting direction to investigate the potential of NGAL/MMP-9 complex in bile in the diagnosis of CCA.

Otherwise, in addition to demonstrating that CEAM6 and NGAL are highly expressed in malignant biliary stenosis compared to benign biliary disease, Lukic et al. found that olfactomedin-4 (OLFM4), syntenin-2 (SDCB2) and Ras-related C3 botulinum toxin substrate 1 (RAC1) were also highly expressed in bile in patients with malignant biliary stenosis, but they did not assess for its diagnostic ability [98]. OLFM4 and RAC1 are also candidate biomarkers for a variety of tumors [115,116].

#### 3.2.4. AAT

Alpha-1-antitrypsin (AAT) is a glycoprotein produced by liver, which forms counterparts with neutrophil elastase, and the disruption of the balance of counterparts will promote tumor progression [117]. AAT was a significantly overexpressed protein in 147 proteins identified by tandem mass spectrometry in bile proteins from six CCA patients and two non-CCA patients in a study of differentially expressed proteins in bile samples from CCA and non-CCA patients [118]. The immunoblotting validation of 54 CCA bile samples showed a positive rate of 70% for AAT in CCA diagnosis, and AAT also remarkable increased in the tissues and stool of CCA patients, suggesting that bile AAT is one of possible biomarkers for diagnosing CCA [118]. The amount of AAT in bile in the eCCA group was considerably higher than that in the benign biliary tract disease group, according to Son et al.’s quantitative proteomics investigation; however, its feasibility as a diagnostic biomarker for CCA has not been confirmed [119]. Oxidative-1 anti-trypsin (ox-A1AT) in human serum is a modified form of AAT and has been used as an oxidative stress indicator for many diseases, which is also one of the candidate biomarkers for screening for opisthorchiasis-associated CCA; this research also supports the close relationship of CCA and AAT [120]. These results approved the potential of AAT to become a CCA diagnostic biomarker, but it needs more clinical data.

#### 3.2.5. S100A8

Calprotectin, a member of the calcium-binding protein S100 leukocyte protein family, is an emerging clinical marker of inflammatory bowel disease [121]. S100A8 is a calcium- and zinc-binding protein released by neutrophils under inflammatory conditions [122]. The two-dimensional strong cation-exchange and reversed-phase nano-scale liquid chromatography–electrospray ionization tandem mass spectrometry analysis (2D-Nano-LC–ESI–MS–MS) of bile in eight patients with benign and malignant gallbladder disease found that the S100A8 of the S100 calcium-binding protein family was elevated in bile in malignant gallbladder cancer, consistent with the trend of S100A8 in tissues [123]. Alternatively, S100A8 plays a promoting role in the metastasis of CCA and is also a prognostic biomarker for breast and bladder cancer [124,125,126]. S100A9, another member of the S100 calcium-binding protein family, is also a candidate biomarker for the diagnosis of CCA, although these studies primarily evaluated S100A9 in serum and tissues [127,128]. Meanwhile, S100A9 is also one of the diagnostic or prognostic candidate biomarkers for a variety of diseases, for example, gastric, bladder and hepatocellular carcinoma [126,129,130,131]. Moreover, it was found that the S100A8/A9 complex can be used for the diagnosis or the evaluation of treatment and prognostic of colorectal cancer and Hodgkin’s lymphoma, which can also promote tumor cell apoptosis under inflammatory conditions [122,132,133]. These data indicate that the bile S100 calcium-binding family monomers or complexes of the proteins may also be a novel diagnostic biomarker of CCA, but it requires further study.

#### 3.2.6. SSP411

SSP411 is a protein that regulates sperm maturation, fertilization and embryo development [134]. The differential protein analysis of 15 samples of CCA patient bile and 10 samples of cholangitis patient bile using two-dimensional electrophoresis and MS/MS analysis, and five different proteins of PGAM1, PDIA3, HSPD1, SSP411 and APOM were randomly selected and validated in bile and tissues; it was found that they had had the same expression trends in bile and tissues [135]. However, only the diagnostic potential of SSP411 in the serum was verified, the diagnostic potential of these five bile proteins was not validated.

#### 3.2.7. TPD52 and DNAJB1

Tumor protein D52-like 2 (TPD52) and DNAJB1 are two carcinogenesis proteins. TPD52L2, a member of the TPD52 family, can be used to predict the prognosis of lung adenocarcinoma, and DNAJB1 is also one of the diagnostic and prognostic markers of pancreatic cancer [136,137]. The isobaric tags for relative and absolute quantitation (iTRAQ) quantitative proteomics analysis of CCA cell lines (TFK1 and HuCCT1) and normal bile duct epithelial cell lines (HiBECs) secreted proteins and validation in tissues and bile and revealed that bile TPD52 and DNAJB1 are potential markers for the diagnosis and predicting prognosis of CCA; nevertheless, the sample size was too little to verify [138].

#### 3.2.8. Others

Opisthorchis viverrini infection is also one of the factors causing CCA [139]. According to the study of Aksorn et al., immunoglobulin heavy chain, translocated in liposarcoma (TLS), Visual System Homeobox 2 (VSX2) and an unnamed protein product, can be used to distinguish the CCA associated with liver fluke infection, but their diagnostic performance is not elaborated [140].

One study has proposed a new method for diagnosing CCA, which is to detect protein pattern analysis instead of looking for individual proteins. Based on this model, they proposed a diagnostic model using 22 bile polypeptides to jointly identify early CCA, with a diagnostic AUC value up to 100% [91]. Urine proteomics can also be used to explore the biomarker distinguishing CCA from benign biliary disease [141]. Voigtlander et al. have built a combined model of bile and urinary proteomics, which have a higher sensitivity and specificity to diagnose CCA than a single model [142]. Then, Voigtlander et al. investigate the interaction of biomarkers with the physiopathological mechanism of CCA, providing a new research direction in which diagnosis biomarkers can be used to discuss the pathogenesis of CCA [17].

Using proteomics to search for bile diagnostic biomarkers for CCA is a recognized method and is also important for the early diagnosis and early treatment of CCA. However, there are two problems: (1) The dry weight of protein in bile accounts for less than 5%, and the currently extracted bile protein contains impurities and a low recovery rate, which will affect the outcome of bile protein identification [143,144]. Therefore, optimizing the bile protein extraction method is one of the first questions that need to be solved. (2) It is necessary to continuously expand the sample size to verify the identified potential protein biomarkers.

### 3.3. Diagnosis Biomarkers in Human Bile Proteins Based on Glycoproteomics

More and more researchers have paid attention to the role of abnormal glycosylation in tumors, such as alpha-fetoprotein (AFP), which is a clinically commonly used diagnosis of liver cancer. Glycoproteomics usually uses proteomics strategies to identify protein and their glycosylation mode and sites. Later, some researchers tried to search for biomarkers of diseases by glycoproteomics.

Wisteria floribunda agglutinin (WFA) is a probe that can specifically identify anti-sialylated mucin 1 (MUC1), and the levels of MUC1 in bile captured by WAF differ greatly between CCA patients and patients with benign biliary disease, so MUC1 in bile is promising for diagnosing CCA, but its specificity is low. Based on the above results, the glycoprotein mass spectrometry of CCA and hepatobiliary stone with the help of WFA found that L1 cell adhesion molecule (L1CAM) had the highest specificity but low sensitivity, and the ability to diagnosis CCA was stronger when two were combined [145,146]. Yamaguchi et al. showed a better diagnostic sensitivity of MUC1 in bile than in cytology, and the combination of bile MUC1 and serum CA19-9 had a higher diagnostic specificity for CCA [147].

MUC5AC is a serum candidate biomarker for the diagnosis of biliary tract cancer [148], and MUC4 is also an independent factor for the evaluation of iCCA and eCCA [149,150]. The diagnostic sensitivity of MUC4 in bile is 27%, and although the sensitivity of combined serum MUC5AC in the diagnosis of biliary tract cancer can be increased to 58%, it is still too low to be used as a diagnostic biomarker for biliary tract cancer [151].

In a nutshell, we identified proteins in human bile that could be used as diagnostic biomarkers for CCA and estimated their feasibility based on diagnostic performance and sample size.

### 3.4. Detection of Protein Biomarkers in Bile

Currently, the commonly used protein detection techniques are ELISA and mass spectrometry, but mass spectrometry is not suitable for clinical routine use due to its high price and complex technology, while ELISA is a common method in clinical practice due to its simplicity [152,153]. Taking the detection of some MMP family members as an example, the detection methods of MMP-7 include ELISA and biosensors with different principles, such as electrochemical biosensors based on enzyme-catalyzed reactions, sensors based on fluorescence resonance energy transfer. However, it cannot be used for quantitative detection because it is susceptible to MMP-2 [153,154,155].

## 4. Search for CCA Diagnosis Biomarkers in Human Bile at Metabolic Compounds Level

The metabolic characteristics of bile are also one of the ways to find diagnostic biomarkers of CCA. Metabolomics refers to a high-throughput analytical chemical method based on mass spectrometry or nuclear magnetic resonance spectroscopy. In recent years, metabolomics has been widely applied to studying the physiopathology, diagnosis and therapy of hepatobiliary diseases, and many achievements have been made [156]. Some studies use metabonomics to find biomarkers for diagnosis of CCA.

Bile acid is an important component of bile, with higher levels of primary bile acid and lower levels of secondary bile acid in CCA compared to benign biliary diseases, which may be associated with bile duct obstruction [157,158,159]. Glycocholic acid (GCA) and taurochenodeoxycholic acid (TCDCA) in bile are candidate surface biomarkers to distinguish CCA from benign biliary disease and pancreatic cancer, where GCA is higher and TCDCA is lower in CCA bile [160].

Hashim et al. analyzed the 1H NMR spectra of bile from 8 patients with CCA and 21 from patients with benign biliary tract disease, showing that the level of bile phosphatidylcholine (PtC) is reduced in CCA, opposite to it is taurine-conjugated bile acids and glycine-conjugated bile acids [14]. The 1H NMR and 31P NMR spectrum of bile also supported the notion that PtC level is low in CCA bile [157,158,161]. Furthermore, Sharif et al. constructed a diagnostic model consisting of PtC, H-18 bile acids and taurine-conjugated bile acids to distinguish CCA from non-malignant biliary disease with a sensitivity of 80% and a specificity of 95% [157]. Albiin et al. also brought a diagnostic model consisting of PtC, bile acid, lipid and cholesterol. This model has a sensitivity of 88.9%, specificity of 87.1% and accuracy of 87.8% [158]. These results suggest that a bile metabolites combined model is one approach to diagnose CCA in the future.

Although lipidomics has been a branch of omics independent of metabolomics, lipids in bile are also a metabolite, so lipidomics is attributed to metabolomics in this review. Lipid oxidation is involved in cell apoptosis, inflammation and oxidative stress, the physiological and pathological processes of tumors, and is also a symbol of tumors [162]. Navaneethan et al. performed a lipidomic analysis of oxidized phospholipid components in bile from patients with benign and malignant bile duct stricture, finding that the accuracy of the combined diagnosis of 1-palmitoyl-2-(9-oxononanoyl)-sn-glycero-3-phosphatidylcholine (ON-PC) and 1-palmitoyl-2-succinoyl-sn-glycero-3-phosphatidylcholine (S-PC) was 91%, much higher than the diagnosis alone [15].

Biliary tract is a bridge to transport bile, and CCA cells metabolites will exist in bile. It is feasible to find diagnostic biomarkers in bile for the non-invasive diagnosis of CCA. Most of the diagnostic biomarkers proposed by bile metabolomics studies are combined diagnoses; compared with a single metabolite diagnosis, joint diagnosis can reduce the clinical diagnostic error caused by individual differences. In addition, this diagnosis method may also be combined with scientific calculation methods to enhance diagnostic performance.

Current diagnostic methods of bile metabolites have their limitations. The chromatographic results is difficult to replicate [163]. Non-chromatographic methods, such as enzyme analysis, which is currently quantitative measurement of TBA clinically, however, can only detect serum samples, and there is no product for bile samples [164]. In contrast to enzyme analysis, NMR can identify metabolites in bile, but they cannot be quantitatively analyzed [165]. Therefore, the current problem is how to achieve the quantitative and qualitative detection of bile metabolites simultaneously.

## 5. Search for CCA Diagnosis Biomarkers in Human Bile at the Extracellular Vesicle Level

The extracellular vesicle (EV) is derived from multivesicular bodies or plasma membrane and carry RNA species, as well as proteins, amplified gene sequences and lipids. Although their physiologic role is currently unknown, EVs are released into the peri-cellular environment, as well as bodily fluids and circulating blood and, as such, may transmit signals from donor to recipient cells. Serum EV can be used to search for the protein biomarker for the diagnosis of CCA [166]. The concentration of EV in bile can also distinguish between malignant biliary stenosis, and its accuracy can be as high as 100% [167].

### 5.1. RNA Diagnostic Biomarker of CCA in EV

#### 5.1.1. MiRNA

Both miR-10a-5p and miR-181a-5 were overexpressed in CCA bile, and miR-10a-5p was also associated with CCA growth [168,169]. However, this study only used two sample sizes, and whether they can be used as CCA diagnostic biomarkers needs to be investigated further.

The type and quantity of miRNAs can be detected in bile EV by qRT-PCR miRNA array technology. Based on this technique, a study constructed a 5-miRNA panel using MOCA calculations for the diagnosis of CCA with a specificity up to 96% [170]. The flaws are also apparent, including small sample size and low sensitivity.

#### 5.1.2. Circ RNA

Microarray technology has the characteristics of high throughput and low consumption. Xu et al. identified circRNA expression profiles in bile derived EV from CCA and patients with benign biliary disease by circRNA microarrays to find a differentially expressed circRNA and named circRNA1(CIRC-CCAC1). CircRNA1 is involved not only in the growth and spread of CCA but also in the differentiation of CCA from benign biliary strictures [171]. Compared to omics technology, it has advantage in simultaneous diagnosis of large numbers of samples to save time and lower costs, which is also a new way of combining medicine and industry.

#### 5.1.3. LncRNA

LncRNA is a non-protein-coding long-chain RNA that can regulate the proliferation, migration, invasion and anti-apoptosis of CCA [172]. The sequencing and differential analysis of bile EV lncRNA from CCA and benign biliary disease revealed two unreported lncRNAs, ENST00000517758.1 and ENST00000588480.1, with low specificity and a medium accuracy rates of diagnosis CCA, although combined diagnosis improved diagnostic sensitivity, the accuracy remains is still not high [173]. Although they cannot use in clinical diagnosis now, this research is the first to verify the diagnostic value of lncRNA in bile, which can provide new directions for future research on lncRNA in the bile of CCA.

### 5.2. CCA Diagnostic Biomarker of Protein in EV

Claudin-3(CLDN3) has an important cell-to-cell tight-junction function, while tumor development and metastasis are thought to involve with the loss of tight-junction function [174]. Proteins in the bile EV of 20 patients with benign and malignant gallbladder diseases were determined by Proteomics analysis, in order to evaluate the expression of CLDN3, showing that CLDN3 is a candidate diagnosis of CCA with an AUC of 94.5% [175].

In recent years, as research into liquid EV has progressed, the importance and value of bile EV in hepatobiliary system illnesses has steadily become apparent. The study in this part confirms the diagnostic value of bile EVs in CCA from the RNA to the protein level, though there is a long way to use in clinical settings.

## 6. Search for CCA Diagnosis Biomarkers in Human Bile at Volatile Organic Compounds Level

Volatile organic compounds (VOS) refer to the headspace gas formed by vaporizing the volatile components in the sample when the sample is placed in a sealed vial and heated to 40 °C. Navaneethan et al. found that VOS can be used to diagnose pancreatic cancer in an early study [176]. In order to expand the role of VOC in the diagnosis of CCA, they collected the bile VOC from 11 patients with CCA and 21 PSC without CCA and analyzed the composition of bile VOS by an ion flow tube mass spectrometry instrument. It was found that specific patterns of formation of acrylonitrile, methyl hexane and benzene could be used to distinguish between CCA and PSC [16]. The disadvantage of VOS technology is its high price. However, if there is an opportunity to transform the diagnosis of bile VOS into respiratory gas, it will be a major advance in terms of saving costs and non-invasive diagnosis.

## 7. Conclusions and Future Perspectives

The bile duct carcinoma of early diagnosis and accurate non-invasive diagnosis helps prolong survival for patients with CCA. In the process of CCA, bile is in direct contact with the lesion, which is one of the candidate specimens when looking for diagnostic biomarkers for CCA. More research methodologies and specimens are being explored to uncover diagnostic or prognostic indicators or therapeutic targets for CCA, such as miRNA, protein and metabolic chemicals in bile, bile EV and VOS. We outline the current state of CCA diagnostic biomarkers and their detection ways in the hopes of generating novel ideas for CCA early detection and non-invasive diagnosis (Table 1). In addition, bile studies can also be used to explore the pathophysiology of CCA [177]. We show the molecular relationship between reported biomarkers in Figure 2, which has huge potential for the parsing mechanism of CCA.

At present, serum CA19-9, with a low diagnostic efficiency, is the main biomarker of body fluids used in clinical diagnosis, and no bile biomarker can be directly used in clinical diagnosis. Since CCA is a relatively low incidence tumor, investigators usually rely on retrospective studies of existing specimens to screen for markers. Therefore, they need to be validated and evaluated in prospective clinical studies in large populations before they have a chance to be used in the clinic.

## Figures and Tables

**Figure 1 cancers-14-03921-f001:**
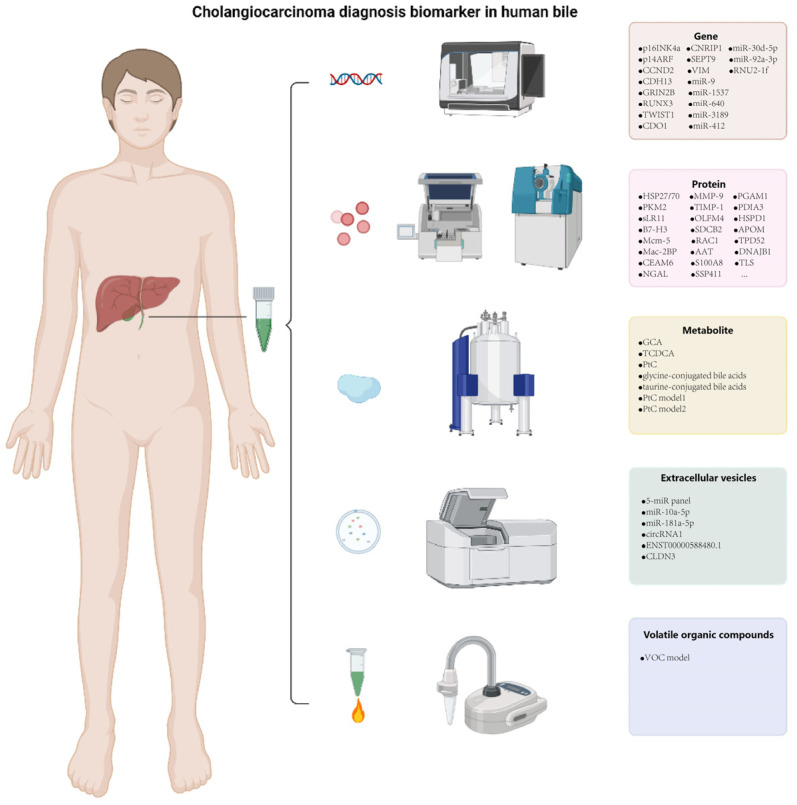
Cholangiocarcinoma diagnosis biomarker in human bile. Classification and identification methods of diagnostic biomarkers for cholangiocarcinoma in human bile. Created with biorender.com.

**Figure 2 cancers-14-03921-f002:**
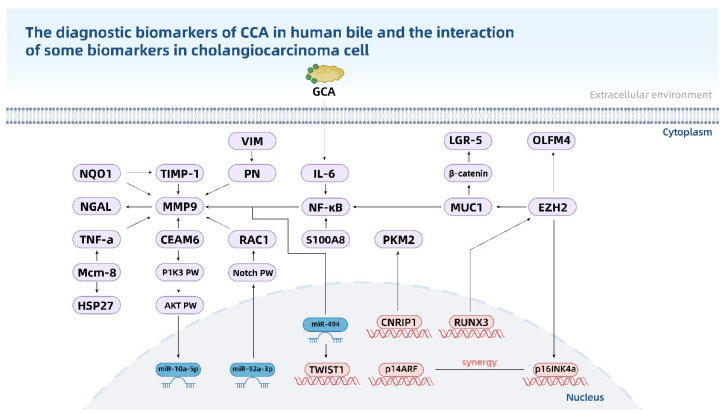
The diagnostic biomarkers of CCA in human bile and the interaction of some biomarkers in cholangiocarcinoma cell. “PW” means pathway.

**Table 1 cancers-14-03921-t001:** Potential diagnosis biomarkers in human bile with cholangiocarcinoma.

Type	Name	Sensitivity	Specificity	AUC	SS	Ref	Remark
Gene	p16INK4a↑	53.6%	93.8%	\	7	[8]	
	p14ARF↑	46.2%	96.9%	\	7	[8]	
	CCND2↑	75.6%	100.0%	\	125	[9]	CD
	CDH13↑	75.6%	100.0%	\	125	[9]	CD
	GRIN2B↑	75.6%	100.0%	\	125	[9]	CD
	RUNX3↑	75.6%	100.0%	\	125	[9]	CD
	TWIST1↑	75.6%	100.0%	\	125	[9]	CD
	CDO1↑	100.0%	90.0%	\	344	[10]	CD
	CNRIP1↑	100.0%	90.0%	\	344	[10]	CD
	SEPT9↑	100.0%	90.0%	\	344	[10]	CD
	VIM↑	100.0%	90.0%	\	344	[10]	CD
	miR-9↑	88.9%	100.0%	97.5%	18	[46]	
	miR-1537↑	67.0%	90.0%	78.0%	83	[54]	
	miR-640↑	50.0%	92.0%	81.0%	83	[54]	
	miR-3189↑	67.0%	89.0%	80.0%	83	[54]	
	miR-412↑	50.0%	89.0%	81.0%	83	[54]	
	RNU2-1f↑	67.0%	91.0%	85.6%	34	[60]	
	miR-30d-5p↑	81.1%	60.5%	73.0%	106	[66]	
	miR-92a-3p↑	65.7%	66.7%	65.2%	106	[66]	
Protein	HSP27↑	90.0%	90.0%	86.0%	20	[13]	
	HSP70↑	80.0%	80.0%	80.5%	20	[13]	
	PKM2↑	52.9%	94.1%	77.0%	74	[11]	
	sLR11↑	100.0%	80.0%	89.0%	147	[82]	
	B7-H3↑	81.7%	69.1%	83.7%	213	[84]	
	B7-H3 + ETFB	88.1%	100.0%	93.9%	213	[84]	CD
	Mcm-5↑	66.0%	94.0%	80.0%	106	[12]	
	Mac-2BP↑	69.0%	67.0%	70.0%	78	[95]	
	Mac-2BP + serum CA19-9	\	\	75.0%	78	[95]	CD
	CEAM6↑	83.0%	93.0%	92.0%	41	[97]	
	CEAM6 + serum CA19-9	\	\	96.0%	41	[97]	CD
	CEAM6↑	87.5%	69.1%	74.0%	83	[98]	
	NGAL↑	94.0%	55.0%	76.0%	59	[101]	
	NGAL↑	77.3%	72.2%	74.0%	40	[102]	
	MMP-9↑	93.9%	32.5%	50.4%	113	[111]	
	TIMP-1↓	96.9%	36.2%	53.9%	113	[111]	
	OLFM4↑	\	\	\	13	[98]	
	SDCB2↑	\	\	\	13	[98]	
	RAC1↑	\	\	\	13	[98]	
	AAT↑	70.0%	\	\	54	[118]	
	S100A8↑	\	\	\	8	[123]	
	SSP411↑	\	\	\	25	[135]	
	PGAM1↑	\	\	\	25	[135]	
	PDIA3↑	\	\	\	25	[135]	
	HSPD1↑	\	\	\	25	[135]	
	APOM↓	\	\	\	25	[135]	
	TPD52↑	\	\	\	37	[138]	
	DNAJB1↑	\	\	\	37	[138]	
	immunoglobulin heavy chain↑	\	\	\	43	[140]	
	TLS↑	\	\	\	43	[140]	
	VSX2↑	\	\	\	43	[140]	
	22 peptides model	\	\	100.0%	107	[91]	
	MUC1↑	90.0%	72.0%	85.0%	58	[145]	
	L1CAM↑	66.0%	93.0%	82.0%	58	[145]	
	MUC1 + L1CAM	90.0%	79.0%	93.0%	58	[145]	CD
	MUC4↑	27.0%	99.0%	\	62	[151]	
	MUC4 + serum MUC5AC	58.0%	87.0%	\	61	[151]	CD
Metabolite	GCA↑	\	\	\	104	[160]	
	TCDCA↓	\	\	\	104	[160]	
	PtC↓	\	\	\	29	[14]	
	glycine-conjugated bile acids↑	\	\	\	29	[14]	
	taurine-conjugated bile acids↑	\	\	\	29	[14]	
	PtC↓	\	\	\	25	[161]	
	PtC model 1↓	80.0%	95.0%	\	25	[157]	
	PtC model2↓	88.9%	87.1%	87.8%	49	[158]	
	ON-PC↑	85.7%	80.3%	86.0%	46	[15]	
	ON-PC + S-PC	100.0%	83.3%	91.0%	46	[15]	CD
EVs source	5-miR panel	67.0%	96.0%	\	6	[170]	
	miR-10a-5p↑	\	\	\	2	[168]	
	miR-181a-5p↑	\	\	\	2	[168]	
	circRNA1↑	\	\	85.7%	84	[171]	
	ENST00000588480.1↑	62.9%	73.2%	68.0%	91	[173]	
	ENST00000517758.1 + ENST00000588480.1	82.9%	\	70.9%	91	[173]	CD
	CLDN3↑	87.5%	87.5%	94.5%	20	[175]	
VOS source	VOC model	90.5%	72.7%	89.0%	32	[16]	

AUC: area under the curve, SS: sample size, Ref: reference, CD: combined diagnosis, EVs: extracellular vesicle, VOS: volatile organic compounds. “↑” in the name means this gene, protein or metabolite is overexpressed in CCA bile; and “↓” in the name means this gene, protein or metabolite is underexpressed in CCA bile. “\” in the name means this value has not been reported in the cited literature.

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
