# Peer review of "Diagnosis Biomarkers of Cholangiocarcinoma in Human Bile: An Evidence-Based Study"

_cancers, 2022, doi:10.3390/cancers14163921_

Round 1

Reviewer 1 Report

This is an excellent and timely topic summarizing research studies looking at biomarkers for cholangiocarcinoma in bile.  The paper would be strengthened by adding a summary clinical paragraph addressing: Are any of these biomarkers currently used as standard of care in diagnosing CCA, and  and when they might be used clinically.

They should also add to the discussion that "bili-seq" NGS genetic analysis (looking at certain gene mutations) on tissue biopsy at ERCP is now in wide use to confirm CCA if brushings are not diagnostic.  They should discuss and cite Singhi AD, et al, Integrating next-generation sequencing to endoscopic retrograde cholangiopancreatography (ERCP)-obtained biliary specimens improves the detection and management of patients with malignant bile duct strictures. Gut. 2020; 69:52-61.

Author Response

Dear reviewer:

We appreciate your kind and instructive comments for our submitted manuscript (No.: cancers-1819427, Title: "Diagnosis Biomarkers of Cholangiocarcinoma in Human Bile: An Evidence-Based Study"). Those comments concerning our manuscript are valuable and helpful for revising and improving our manuscript, as well as the important guiding significance to our works. We have studied comments carefully and revised our manuscript point-by-point. We hope to meet with your approval. The main corrections in the paper and the responses to the reviewer’s comments are appended below, which are also marked up using the “Track Changes” function in the manuscript.

Point 1: Are any of these biomarkers currently used as standard of care in diagnosing CCA, and when they might be used clinically?

Response 1: The most widely used biomarker clinically is serum CA19-9, but if has low sensitivity and specificity. But no biomarkers in bile have been used for clinical diagnosis of cholangiocarcinoma. So, we write this review to summary potential biomarker in bile. We added in the manuscript how can these biomarkers be used clinically and hope more researchers will conduct prospective studies to evaluate their performance in diagnosing CCA. We have added the following (P14, Line592-597):

“At present, serum CA19-9, with low diagnostic efficiency, is the main biomarker of body fluids used in clinical diagnosis, and no bile biomarker can be directly used in clinical diagnosis. Since CCA is a relatively low incidence tumor, investigators usually rely on retrospective studies of existing specimens to screen for markers. Therefore, they need to be validated and evaluated in prospective clinical studies in large populations before they have a chance to be used in the clinic.”

Point 2: You should also add to the discussion that "bili-seq" NGS genetic analysis (looking at certain gene mutations) on tissue biopsy at ERCP is now in wide use to confirm CCA if brushings are not diagnostic. You should discuss and cite Singhi AD, et al, Integrating next-generation sequencing to endoscopic retrograde cholangiopancreatography (ERCP)-obtained biliary specimens improves the detection and management of patients with malignant bile duct strictures. Gut. 2020; 69:52-61.

Response 2: Thanks for your suggestion. According to your comment, we have added the new diagnostic methods described in “Integrating next-generation sequencing to endoscopic retrograde cholangiopancreatography (ERCP)-obtained biliary specimens improves the detection and management of patients with malignant bile duct stricturesin. ” and cited it in the revised manuscript as follows (P2, Line47-50).

“Singhi et al. proposed a multigene a next-generation sequencing (NGS) diagnosis method in tissues, which was significantly more effective than serum CA19-9 and pathological evaluation. But it has not been included in the management guidelines of CCA [7].”

Reviewer 2 Report

This paper well elucidated the role of bile in the diagnosis of cholangiocarcinoma and systematically summarized known biomarkers.

I belive It will provides a lot of useful information to researchers working in this field.

I would like to suggest a few issues. 

The authors approached the biomarkers identified for each genome and proteomic in detail.

It would be nice if you could provide some more information about the macro perspective below.

1. Each paper used various method for bile acqusition, preparation and anlylsis method.  Could you summarize and introduce breifly about them?

2. Bile can be used as method of liquid biopsy. As authors described, bile may be ideal  fluid for cholangiocarcinoma,  because it contact to cancer directly. But is bile really better one for liquid biopsy than blood?  Is diagnostic value of bile comparable to that of tumor tissue?  If there are papers comparingbile to other body fluid (blood) or tissue, please summarize.   (for example : Cancers (Basel) . 2021 Sep 12;13(18):4581)

Author Response

Dear reviewer:

  We appreciate your kind and instructive comments for our submitted manuscript (No. : cancers-1819427, Title: "Diagnosis Biomarkers of Cholangiocarcinoma in Human Bile: An Evidence-Based Study"). Those comments concerning our manuscript are valuable and helpful for revising and improving our manuscript, as well as the important guiding significance to our works. We have studied comments carefully and revised our manuscript point-by-point. We hope to meet with your approval. The main corrections in the paper and the responses to the reviewer’s comments are appended below, which are also marked up using the “Track Changes” function in the manuscript.

Point 1: Each paper used various methods for bile acquisition, preparation and analysis method. Could you summarize and introduce them briefly? 

Response 1: Thanks for your suggestion. We have added a paragraph to describe the methods of bile acquisition, storage and analysis briefly as follows (P2, Line51-71).

“Bile secreted by the liver cells, generally stored in the gall bladder, discharged to the duodenum in the digestive process, is an important part of the biliary system. At present, the main methods to obtain bile are duodenal drainage, gallbladder puncture and direct surgery, among which surgical methods include ERCP and cholecystectomy. Moreover, the bile collected by duodenal drainage contains part of gastric juice and pancreatic juice, which is not conducive to the purification and analysis of bile. Bile samples should be transported on ice after acquisition, and frozen at −80°C until use. For different types of bile candidate markers, treatment and analysis methods are as follows: 1) Gene level: After extracting DNA from the bile, amplification of the target fragment by PCR to obtain the expression of potential diagnostic biomarkers and analysis of data [8]. For RNA, target RNA reverse transcription is required to form a stable CDNA before PCR amplification and data analysis, real-time quantitative PCR is usually used [9,10]. 2) Protein level: ELISA or automatic immunofluorescence analysis is adopted to verify suspicious protein biomarkers in bile, but when exploring unknown diagnostic biomarkers, proteomics methods are being generally used to find markers and then checked by ELISA in bile [11-13]. 3) Metabolism level: For bile metabolites, the diagnostic efficacy of suspected biomarkers is generally analyzed by magnetic resonance spectroscopy or quantitated by liquid chromatography tandem mass spectrometry [14,15]. 4) VOC is mainly through heating the sealed bottle containing bile and ex-tracting the air above, and the collected gas is analyzed by ion flow tube mass spectrometry, so as to identify the markers and analyze the diagnostic effect [16].”

Point 2: Bile can be used as a method of liquid biopsy. As the authors described, bile may be an ideal fluid for cholangiocarcinoma, because it contacts cancer directly. But is bile really a better one for liquid biopsy than blood? Is the diagnostic value of the bile comparable to that of tumor tissue? If there are papers comparing bile to other body fluid (blood) or tissue, please summarize. (for example: Cancers (Basel) . 2021 Sep 12;13(18):4581)

Response 2: Thanks for your suggestion. We illustrate that bile is better than serum in diagnosing CCA, and its sensitivity is better than ERCP cytology (P2, Line77-79) as follows.

“ The consistency of gene mutations between tumor tissues and bile was twice that in serum. and serum had the lowest levels of ctDNA [19]. Another study also proved that bile ctDNA is superior to ERCP cytology [20] ”
